# *Brassica rapa* Nitrate Transporter 2 (*BrNRT2*) Family Genes, Identification, and Their Potential Functions in Abiotic Stress Tolerance

**DOI:** 10.3390/genes14081564

**Published:** 2023-07-31

**Authors:** Bingcan Lv, Yifan Li, Xiaoyu Wu, Chen Zhu, Yunyun Cao, Qiaohong Duan, Jiabao Huang

**Affiliations:** College of Horticulture Science and Engineering, Shandong Agricultural University, Tai’an 271000, China

**Keywords:** *Brassica rapa*, *NRT2* gene family, abiotic stress, bioinformatics, expression profile

## Abstract

Nitrate transporter 2 (NRT2) proteins play vital roles in both nitrate (NO_3_^−^) uptake and translocation as well as abiotic stress responses in plants. However, little is known about the *NRT2* gene family in *Brassica rapa*. In this study, 14 *NRT2s* were identified in the *B. rapa* genome. The *BrNRT2* family members contain the PLN00028 and MATE_like superfamily domains. *Cis*-element analysis indicated that regulatory elements related to stress responses are abundant in the promoter sequences of *BrNRT2* genes. *BrNRT2.3* expression was increased after drought stress, and *BrNRT2.1* and *BrNRT2.8* expression were significantly upregulated after salt stress. Furthermore, protein interaction predictions suggested that homologs of BrNRT2.3, BrNRT2.1, and BrNRT2.8 in *Arabidopsis thaliana* may interact with the known stress-regulating proteins AtNRT1.1, AtNRT1.5, and AtNRT1.8. In conclusion, we suggest that *BrNRT2.1*, *BrNRT2.3,* and *BrNRT2.8* have the greatest potential for inducing abiotic stress tolerance. Our findings will aid future studies of the biological functions of *BrNRT2* family genes.

## 1. Introduction

Nitrogen is an important element for the growth and development of plants, as it plays a significant role in shaping the composition of proteins and nucleic acids. Furthermore, nitrogen is a key factor affecting both the yield and quality of crops. Within the soil environment, the forms of nitrogen that plants utilize include ammonium nitrogen (NH_4_^+^) and nitrate nitrogen (NO_3_^−^). Nitrate is the primary nitrogen source for most dryland crops, including wheat, soybean, and Chinese cabbage. Its uptake and transport are facilitated by nitrate transport proteins. Nitrate transporter 2 (NRT2) is part of the NITRATE/NITRITE PORTER (NNP) family, which in turn belongs to the MAJOR FACILITATOR SUPERFAMILY (MFS). The structure of the NRT2 protein generally includes 500–600 amino acids (aa) and contains 12 transmembrane helical segments [1]. This protein was first discovered and characterized in *Aspergillus nidulans* [2]. Molecular cloning of the *NRT2* gene in higher plants was first reported in barley [3]. *NRT2* family members are categorized as high-affinity transport system (HATS), and most of the family members need to bind to the molecular chaperone nitrate-assimilation-related 2 (NAR2 or NRT3) to activate their high-affinity transport activity, which is important when there is a low concentration of exogenous nitrate; *NRT2* family members, thus, play a key role in plant growth and development.

Several studies have reported the functions and evolutionary history of *NRT2* genes in different plant species. NRT2s are responsible for nitrate absorption and transport in plants. In *A. thaliana*, seven members of the *NRT2* family have been identified [4]. *AtNRT2.1* performs a key dual role in controlling root development with external NO_3_^−^ availability [5], and *AtNRT2.2* and *AtNRT2.1* synergistically regulate NO_3_^−^ uptake and the dynamic response of plants to changes in the environmental nitrogen content [6]. *AtNRT2.4* plays a dual role in the shoots and roots of nitrogen-starved plants [7]. *AtNRT2.5* is necessary to support the growth of nitrogen-starved mature plants by working with *AtNRT2.1*, *AtNRT2.2*, and *AtNRT2.4*, which ensures the efficient nitrate uptake, and by facilitating nitrate loading into the phloem during nitrate remobilization [8]. *AtNRT2.7* is mainly responsible for the loading of nitrate in seed vacuoles and plays a role in NO_3_^−^ storage [9]. In bread wheat (*Triticum aestivum* L.), *TaNRT2.1* is involved in nitrate uptake at the post-flowering stage [10]. In chrysanthemum (*Chrysanthemum morifolium*), *CmNRT2.1* is a nitrate-inducible gene, and its activity can affect nitrate uptake [11]. In pineapple (*Ananas comosus*), *AcNRT2.2* is highly expressed in the roots, suggesting that *AcNRT2.2* might have a significant impact on alleviating nitrate deficiency [12].

NRT2s also play a role in mediating resistance to biotic and abiotic stress. In *A. thaliana*, *AtNRT2.1* has been shown to regulate root hydraulic conductivity and plasma membrane aquaporin activity [13], suggesting that it may enhance plant drought resistance and other processes related to root hydraulic conductivity. Additionally, *AtNRT2.1* works as a significant contributor to Cd uptake by modulating nitrate uptake in high-affinity nitrate systems [14], and *NRT2.1* can also affect plant disease resistance by downregulating biotic stress defense mechanisms and promoting abiotic stress responses [15]. *AtNRT2.6* expression is induced after inoculation of *A. thaliana* with the phytopathogenic bacterium *Erwinia amylovora*. The reduced expression of *NRT2.6* can reduce pathogen tolerance, and its activity is likely linked to ROS production under biotic and abiotic stress [16]. In *B. napus* (*Brassica napus* L.), a total of seventeen members were identified. *BnNRT2.1a*, *BnNRT2.5s*, and *BnNRT2.7s* were found to be involved in the response to waterlogging stress. *BnNRT2.7s* plays an important role under P- and K-deficient conditions [17]. In tomato (*Solanum lycopersicum*), four *NRT2* members have been identified to positively affect the response of plants to drought and salt stress [18].

In addition, NRT2s also regulate the transport of auxin to the root system of plants to participate in nitrate-dependent root elongation [19]. They also play a role in regulating the control of cytokinins [20] and are involved in the biosynthesis and signal transduction of ethylene [21]. Their significance is further supported by their contribution to the root morphogenesis of *A. thaliana* [5]. They enhance the pH-buffering capacity of plants [22] and promote the uptake of manganese [23] and phosphorus in rice [24].

Chinese cabbage (*B. rapa*) is a biennial herb of the genus *Brassica* in the Cruciferae family that is widely grown in Asia. It thrives in mild climates and soils with abundant moisture and nitrogen fertilizer. However, *B. rapa* exhibits low nitrogen utilization and frequently faces challenges such as drought and salt stress, which significantly hinders its yield and quality [25,26]. An increasing number of studies indicate that processes involved in nitrogen assimilation, metabolism, and transport are intimately tied to drought and salt stress in plants. For example, the adverse effects of drought stress on *Malus prunifolia* can be mitigated when the nitrogen supply is robust [27]. In the case of oil palm, reactive nitrogen metabolic activities and nitrate assimilation processes contribute to the response of plants to drought stress [28]. Additional studies on sorghum and tomato revealed that the introduction of exogenous nitrogen could notably alleviate the intake of Na^+^ and bolster the K^+^ content in these plants [29,30]. There is, thus, a need to clarify the roles and molecular mechanisms of related genes. This knowledge is important for enhancing the yield and quality of *B. rapa* and strengthening its resilience to drought and salt stress. In this study, we obtained basic information on *BrNRT2* genes and their expression profiles after drought stress and salt stress treatments. The insights gained from this study will contribute to future studies aimed at clarifying the functions of *NRT2* genes.

## 2. Materials and Methods

### 2.1. Identification, Physicochemical Characterization, Chromosomal Localization, and Subcellular Localization of BrNRT2s

In this study, the whole genome sequence of *A. thaliana* was downloaded from EnsemblPlants (release 56) (http://plants.ensembl.org/, accessed on 1 May 2023) to extract the NRT2 protein sequence of *A. thaliana* via TBtools (v1.120) [31]. *B. rapa NRT2* (*BrNRT2*) genes were predicted from the *B. rapa* genome via the BLASTP tool with the *A. thaliana NRT2* (*AtNRT2*) aa sequences as input, and candidate genes were then further identified by conserved structural domain analysis. TBtools (v1.120) combined with Expasy (v3.0) (https://www.expasy.org/, accessed on 1 May 2023) [32] was used to analyze the physicochemical properties of the *BrNRT2* genes, such as isoelectric point (pI), protein molecular weight (MW), and amino acid length (aa). Subcellular localization was predicted using WoLF PSORT (v0.2) [33] (http://www.genscript.com/wolf-psort.html, accessed on 1 May 2023). We then extracted the location information of *BrNRT2* family members from the gene structure annotation information file of *B. rapa* and mapped their positions on chromosomes using TBtools (v1.120).

### 2.2. Phylogenetic Tree Construction and Collinearity Analysis

The aa sequences of BrNRT2s, AtNRT2s, OsNRT2s, PtNRT2s, and HvNRT2s were aligned using the MUSCLE algorithm in MEGA X (v11.0.13), and a phylogenetic tree was constructed using the maximum likelihood method (model: LG + G) in MEGA X with 1000 bootstrap replicates to ensure the accuracy and reliability of our results [34]. The phylogenetic tree was visualized using the online software iTOL (v6.8) [35] (https://itol.embl.de/, accessed on 1 May 2023). To explore the genetic homology across different plant species, we downloaded the genomic and annotation information files of Arabidopsis, soybean, and potato from the EnsemblPlants database (release 56) (http://plants.ensembl.org/, accessed on 1 May 2023). The MCScanX (Multiple Collinearity Scan toolkit) plug-in in TBtools (v1.120) was used to analyze interspecies collinearity.

### 2.3. Gene Structure and Conserved Domain Analysis

The structure of *BrNRT2s* was visualized using the Visualize Gene Structure program of TBtools (v1.120) [31].

We analyzed the conserved domains of *BrNRT2s* using the Web CD-Search tools (v3.18) on the NCBI website (https://www.ncbi.nlm.nih.gov/cdd, accessed on 1 May 2023) [36], and we used TBtools (v1.120) to visualize the results. The conserved motifs of the identified *BrNRT2s* were analyzed using the online tool MEME (v5.5.1) (http://meme-suite.org/tools/meme, accessed on 1 May 2023), and the predicted number of motifs was set to 10 [37].

### 2.4. Cis-Regulatory Elements, and GO Enrichment Analysis

TBtools (v1.120) was used to extract the 2.0 kb promoter sequence upstream of the start codon of *BrNRT2s*. The characteristics of the *cis*-regulatory elements were predicted using the PlantCARE (v1) (http:/bioinformatics.psb.ugent.be/webtools/plantcare/html/, accessed on 3 May 2023) [38] with the default parameters.

In addition, to analyze *NRT2s* based on their functional similarity, we performed GO enrichment analysis using the online DAVID database (v2023q1) [39] (https://david.ncifcrf.gov/, accessed on 3 May 2023), and the GO annotation data were processed and graphically demonstrated using bioinformatics (http://www.bioinformatics.com.cn/, accessed on 3 May 2023).

### 2.5. Transcriptome Data Expression, Stress Treatments, Total RNA Extraction, and qRT-PCR

Transcriptome data for the tissue-specific expression of *B. rapa* (GSE43245) and drought-sensitive plants under drought stress (GSE73963) were downloaded from the Brassicaceae Database (v3.0) (http://www.brassicadb.cn/, accessed on 6 May 2023). Heatmaps of gene expression profiles were prepared using TBtools (v1.120).

The first-generation hybrid cultivar of *B. rapa* with stable self-incompatibility was used for stress treatments. Plump seeds were seeded in MS Modified Medium (with vitamins, Sucrose, Agar) (PM10121-307, Coolaber, Beijing, China) and cultivated in a plant incubator (16 h light/8 h dark photoperiod at 25 °C, light intensity 2000 lx). When the seedlings were 4 weeks old, seedlings with similar growth conditions were subjected to stress treatments. These seedlings were immersed in a 150 mmol·L^−1^ NaCl solution prepared with half-strength Hoagland nutrient solution (pH = 5.8) to simulate salt stress; they were also immersed in 15% PEG6000 to simulate drought conditions, whereas normal hydroponic seedlings were used as the control (the hydroponic pots had a volume of 10 L). All stress treatments were administered for 0, 4, 6, and 12 h for a total of 4 treatments. Each treatment consisted of three biological replicates. The leaves and roots of *B. rapa* under drought and salt stress were sampled and used for RNA extraction and qRT-PCR analysis.

Total RNA was extracted using the SteadyPure Universal RNA Extraction Kit (Accuratel Biology, Hunan, China). The qRT-PCR primer sequences (Appendix A) were designed using the qPrimerDB-qPCR Primer Database (v2.0) (https://biodb.swu.edu.cn/qprimerdb/, accessed on 11 May 2023) [40] and synthesized by Qingdao PrimeTech Zixi Biotechnology Co. Ltd. The qRT-PCR reactions were performed on a QuantStudio 3 system (Applied Biosystems, Waltham, MA, USA) with ChamQ SYBR qPCR Master Mix (Vazyme, Nanjing, China), and *BraActin2* (accession: *Bra037560*) was used as the internal reference gene. The reactions were performed with three technical replicates. The relative expression levels of *BrNRT2* genes were analyzed using the 2^−∆∆CT^ method and graphed using Excel 2016.

### 2.6. NRT2 Protein Secondary Structure, Tertiary Structure, Protein Transmembrane Domains, and Protein Interaction Network Analysis

For protein secondary structure analysis, we utilized the PRABI tool (v5.0.5) (https://npsa-prabi.ibcp.fr/cgi-bin/npsa_automat.pl?page=npsa_sopma.html, accessed on 12 May 2023), and the results were analyzed and plotted using Excel 2016. SWISS-MODEL (v1.0) was used to (https://swissmodel.expasy.org/, accessed on 12 May 2023) [41] predict tertiary structure of BrNRT2 proteins.

TMHMM Server v2.0 (https://services.healthtech.dtu.dk/service.php?TMHMM-2.0, accessed on 12 May 2023) was used to predict the transmembrane domains of BrNRT2 proteins [42].

The STRING website (v11.5) (https://cn.string-db.org/, accessed on 12 May 2023) was used to predict protein–protein interaction (PPI) relationships [43]. In this study, we selected “*A. thaliana*” as the focal organism and used the “single protein by sequence” method. Specifically, we searched for *A. thaliana* NRT2 proteins in the STRING database that were similar to BrNRT2.1, BrNRT2.3, and BrNRT2.8 aa sequences using the BLASTP method. From this analysis, we selected the most similar proteins to establish a comprehensive protein interaction network diagram.

## 3. Results

### 3.1. Identification, Physicochemical Characterization, Chromosomal Localization, and Subcellular Localization of BrNRT2s

A total of 14 *BrNRT2* genes were screened from the *B. rapa* genome, and they were unequally distributed on 6 of the 10 chromosomes of *B. rapa* (Figure 1). There were two *BrNRT2* genes on chromosomes 1, 2, 8, 9, and 10. Four genes were mapped to chromosome 6. They were named *BrNRT2.1~BrNRT2.14* (Table 1) according to their sequences on the chromosomes. To investigate the physicochemical properties of these *BrNRT2* genes, we used ExPASy online software and found that the isoelectric point (pI) of the proteins ranged from 6.89 (*BrNRT2.7*) to 9.15 (*BrNRT2.4*). The number of basic proteins (12) was significantly greater than the number of acidic proteins (2). The molecular weights of the proteins ranged from 52,132.17 Da (*BrNRT2.3*) to 62,619.10 Da (*BrNRT2.9*). The lengths of the BrNRT2 proteins ranged from 484 aa (*BrNRT2.3*) to 575 aa (*BrNRT2.9*). The subcellular localization analysis of *BrNRT2s* showed that all BrNRT2 proteins were distributed in the plasma membrane (plas), suggesting that these proteins might perform their functions on the plasma membrane and that they potentially mediate nitrate transport.

### 3.2. Phylogenetic Tree Construction and Collinearity Analysis

To better understand the evolutionary features among *NRT2* family members, a phylogenetic tree based on sequence similarity was created for the *NRT2* family genes from diverse plant species using the maximum likelihood method (Figure 2A). *BrNRT2* genes were classified into four subfamilies: *BrNRT2.1*, *BrNRT2.2*, *BrNRT2.4*, *BrNRT2.7*, *BrNRT2.8*, and *BrNRT2.14* belong to the same subfamily; *BrNRT2.5*, *BrNRT2.6*, *BrNRT2.10*, *BrNRT2.11*, *BrNRT2.12*, and *BrNRT2.13* belong to the same subfamily; *BrNRT2.3* belongs to a separate subfamily; and *BrNRT2.9* belongs to another subfamily. *BrNRT2.3* and *AtNRT2.7, BrNRT2.9* and *AtNRT2.5*, and *BrNRT2.13* and *AtNRT2.4* are directly homologous. Most *BrNRT2* genes were found to have a covalent relationship with *AtNRT2* genes.

To gain further insights into the origin and evolution of *BrNRT2* genes, we examined the syntenic relationships between *B. rapa* and three other plants (*A. thaliana*, *G. max,* and *S. tuberosum*). There were nine collinear pairs of *BrNRT2* genes and *AtNRT2* genes. There were six collinear pairs of *BrNRT2* genes and *GmNRT2* genes and six collinear pairs of *BrNRT2* genes and *StNRT2* genes. These observations suggest that *B. rapa* and *A. thaliana* are evolutionarily closely related and functionally similar; these findings aid exploration of the functions of *BrNRT2s* (Figure 2B).

### 3.3. Gene Structure and Conserved Domain Analysis

To explore the structure and function of *BrNRT2* genes, we visualized and analyzed the exon–intron structure of *BrNRT2* genes. The gene structure analysis (Figure 3A) showed that *BrNRT2.1*, *BrNRT2.3*, and *BrNRT2.8* had one intron and two exons. *BrNRT2.10*, *BrNRT2.11*, *BrNRT2.12*, and *BrNRT2.14* had two introns and three exons. *BrNRT2.2*, *BrNRT2.4*, *BrNRT2.5*, *BrNRT2.6*, *BrNRT2.7*, and *BrNRT2.13* had three introns and four exons. The gene structure of *BrNRT2.9* is significantly different from other family members, with 6 introns and 7 exons.

To predict the conserved domains of NRT2 proteins in *B. rapa* (Figure 3B), we utilized NCBI-CDD and identified two functional domains present in this gene family: PLN00028 and MATE_like superfamily. *BrNRT2.9* has a unique combination of PLN00028 and MATE_Like superfamily domains, suggesting that this gene potentially has special functions. The other 13 family members only have PLN00028 domain.

We predicted 10 conserved motifs of *BrNRT2* genes (Figure 3C,D). All *BrNRT2* genes contained Motif 1, Motif 2, Motif 4, and Motif 5, indicating that these are the core motifs of *BrNRT2s*. *BrNRT2.9* contained a duplicate Motif 8. *BrNRT2.1*, *BrNRT2.2*, *BrNRT2.7*, *BrNRT2.8*, *BrNRT2.10*, *BrNRT2.12*, *BrNRT2.13*, and *BrNRT2.14* all had 10 Motifs; Motif 7 and Motif 9 were absent in *BrNRT2.4*, Motif 8 was absent in *BrNRT2.5*, Motif 10 was absent in *BrNRT2.3* and *BrNRT2.11*, and Motif 6 was absent in *BrNRT2.6*.

### 3.4. Cis-Elements Analysis

To further investigate the mechanism of the *BrNRT2* genes in response to abiotic stress, the 2 kb promoter sequences upstream of the start codon of *BrNRT2s* were analyzed (Figure 4). Each member of the *BrNRT2* family contains abundant light-responsive elements, which emphasizes the key role of light signal regulation in the growth and development of *B. rapa*. Additionally, all members contain phytohormone response elements; approximately 90% of the *BrNRT2* genes contain ABA elements, and approximately 85% of *BrNRT2* genes contain methyl jasmonate (MeJA)-responsive elements. With the exception of *BrNRT2.7* and *BrNRT2.12*, all members contained *cis*-regulatory elements related to abiotic stress, including low-temperature-responsive elements, drought-inducibility elements, defense- and stress-responsive elements, and anaerobic induction elements. These results indicate that members of the *BrNRT2* gene family play a fundamental role in enhancing abiotic stress tolerance in plants.

### 3.5. GO Enrichment Analysis

GO analysis on the *BrNRT2* genes contributed to determining their functions. GO analysis showed that *BrNRT2* genes were enriched in biological process (BP), cellular component (CC), and molecular function (MF) (Figure 5; Appendix A). Based on a previous study, nitrate transporters play an essential role in drought and salt tolerance [18]. The three enriched terms in the BP category were nitrate transport (GO:0015706), cellular response to nitrate (GO:0071249), and nitrate assimilation (GO:0042128). This prompted us to further characterize the expression changes of *BrNRT2* family members under drought and salt stress conditions. The three enriched terms in the CC category were plant-type vacuole membrane (GO:0009705), plasma membrane (GO:0005886), and integral component of membrane (GO:0016021). The GO–MF enrichment results revealed one enriched term: nitrate transmembrane transporter activity (GO:0015112). Two *BrNRT2* genes were enriched in the nitrate assimilation term; nine genes were enriched in nitrate transport and cellular response to nitrate, albeit with a small *p*-value but high confidence; and nine genes were enriched in plant-type vacuole membrane, plasma membrane, integral component of membrane, and nitrate transmembrane transporter activity.

### 3.6. Gene Expression Analysis

#### 3.6.1. Tissue-Specific Expression

To investigate the expression of the *BrNRT2* genes in various tissues of *B. rapa*, we obtained tissue-specific transcriptome data from the Brassicaceae Database. The transcriptome data of *BrNRT2s* extracted from five different tissues (root, stem, flower, leaf, and silique) (Figure 6; Appendix A) revealed that the expression levels of *BrNRT2.1*, *BrNRT2.5*, *BrNRT2.6*, *BrNRT2.8*, *BrNRT2.10*, *BrNRT2.11*, *BrNRT2.12*, and *BrNRT2.13* were significantly higher in the roots than in the other tissues, indicating that these genes may play an important role in nitrate uptake from the soil and in the response to root-associated stresses. *BrNRT2.4* and *BrNRT2.14* were only highly expressed in the stems. These genes, which were specifically expressed in a single tissue, may play a role in loading and discharging nitrate. *BrNRT2.2* and *BrNRT2.7* were highly expressed in stem and flower tissues. *BrNRT2.3* was highly expressed in stem and leaf tissues, and *BrNRT2.9* was highly expressed in root, stem, and leaf tissues, suggesting that these genes may play a role in the long-distance transport of nitrate from the roots to the aboveground parts. The expression of all genes in the callus was not significant. These tissue-specific expression differences among the *BrNRT2* genes suggest that they play distinct roles in different stages of plant development.

#### 3.6.2. Effect of Drought Stress on the Expression of BrNRT2s

Drought stress can negatively affect *B. rapa* production. To gain a preliminary understanding of the function of *BrNRT2* genes in response to drought stress, we obtained the transcriptome sequencing data of drought-sensitive (DS) *B. rapa* from the Brassicaceae Database (Figure 7A) and used RNA-seq to detect the expression levels of *BrNRT2* genes after drought stress (Figure 7B). Three genes (*BrNRT2.1*, *BrNRT2.3*, and *BrNRT2.10*) with high expression levels in *B. rapa* were selected for further qRT-PCR detection (Figure 7C). The relative expression of *BrNRT2.1* peaked after 4 h of drought treatment and then decreased. The relative expression of *BrNRT2.3* was upregulated significantly at all time points, and its expression was 4–7-fold higher under drought stress relative to the CK treatment. The relative expression of *BrNRT2.10* initially increased and then decreased, reaching a maximum after 4 h of drought treatment.

#### 3.6.3. Effect of Salt Stress on the Expression of BrNRT2s

Salt stress can also negatively affect the yield and quality of *B. rapa*. Based on the results of the RNA-seq assay of the *NRT2* genes in *B. rapa* following exposure to salt stress (Figure 8A), we selected three genes (*BrNRT2.1*, *BrNRT2.3*, and *BrNRT2.8*) with high expression levels in *B. rapa* for qRT-PCR detection (Figure 8B). The relative expression levels of *BrNRT2.1* and *BrNRT2.8* were significantly upregulated at each time point after treatment, and this was consistent with the results of the RNA-seq analysis. The relative expression of *BrNRT2.3* first increased and then decreased, reaching a peak after 4 h of salt treatment. Overall, the pronounced changes in the expression of *NRT2* members under salt stress suggest that there was a relationship between plant responses to salt stress and nitrogen transportation.

### 3.7. Protein Secondary Structure and Tertiary Structure Prediction of BrNRT2s

The structure of proteins is inextricably linked to their biological functions; thus, study of the structure of NRT2 proteins can provide important insights into their functions. In this study, we analyzed the predicted protein secondary structures of BrNRT2s and found that all members had α-helixes, random coil components, extended strands, and β-turns. The α-helixes accounted for the highest proportion of the protein secondary structures of BrNRT2s, and the β-turns accounted for the lowest proportion of protein secondary structures (Figure 9A). The different proportions of protein secondary structures in different family members might be related to the diverse roles of *BrNRT2* genes in cell metabolism. The SWISS-MODEL analysis revealed that members of the same subgroup exhibit high structural similarity in protein tertiary structure (Figure 9B), indicating that they maintained homologous structures during the evolutionary process; this provides basic information that could aid subsequent studies of the functions of BrNRT2 proteins.

### 3.8. Prediction of BrNRT2 Protein Transmembrane Domains

We used TMHMM to predict the transmembrane domains of NRT2 proteins in *B. rapa* (Figure 10). Only the BrNRT2.9 protein contained 13 transmembrane domains, and the remaining 13 BrNRT2 proteins had 8–11 transmembrane domains, suggesting that *BrNRT2.9* can respond more quickly to environmental changes and has a stronger nitrate transport function. Additionally, the number of aa and their positions in each transmembrane domain are similar, indicating that they have similar structures and functions.

### 3.9. Analysis of the Protein Interaction Network of BrNRT2s

Proteins seldom perform their functions alone but interact with other proteins present in their surroundings to accomplish their biological activities [44]. Therefore, understanding protein interactions is necessary for elucidating the mechanisms underlying cellular functions. *B. rapa* and *A. thaliana* are closely related, and the functions of the *AtNRT2* genes have been intensively studied. Therefore, we could predict the function of corresponding homologous genes in *B. rapa* through a PPI analysis of the *AtNRT2* genes, which would help further clarify the functions of *BrNRT2* family members. We found that the relative expression of *BrNRT2.3* was significantly upregulated after drought stress, and the relative expression of *BrNRT2.1* and *BrNRT2.8* was significantly upregulated after salt stress according to analysis of the qRT-PCR data. To clarify the interaction network of BrNRT2 proteins, integrated network maps of the homologous genes of *BrNRT2.1*, *BrNRT2.3*, and *BrNRT2.8* and their interacting proteins were constructed based on the resources and algorithms available in the STRING database to identify their functions as well as physical interactions. The homolog of BrNRT2.3 in *A. thaliana* was AtNRT2.7, and the predicted protein interactions (Figure 11A) revealed strong interactions of AtNRT2.7 with nitrate reductase (NIA), nitrite reductase 1 (NIR1), and nitrate transporter 1 (NRT1). The homolog of BrNRT2.1 and BrNRT2.8 in *A. thaliana* was AtNRT2.6. AtNRT2.6 expresses high-affinity nitrate transporter proteins and sequentially interacts with nitrate reductase 1 (NIA1) to participate in nitrate assimilation; it also strongly interacted with nitrite reductase 1 (NIR1), nitrate transporter 1 (NRT1), and ammonium transporter (AMT) (Figure 11B). Some members of the NRT1 family have been shown to enhance drought and salt stress tolerance in plants. These findings suggest that BrNRT2s may interact with proteins in different families to respond to abiotic stress.

## 4. Discussion

Nitrate is an essential element for plant growth, as it serves as a key nutrient in the nitrogen assimilation pathway and as a vital signal for plant development [45]. *NRT2* family members are categorized as the high-affinity transport system (HATS) and are known to play key roles in nitrate uptake, transport, and response to biotic and abiotic stresses in various plant species. However, the *NRT2* gene family have not been well studied in *B. rapa*. In this study, we analyzed 14 *NRT2* genes from *B. rapa*, including their physicochemical properties, structural features, phylogenetic relationships, *cis*-elements, GO enrichment, expression patterns under abiotic stress, protein structures, and protein interactions. These results aided the analysis of their gene functions.

Phylogenetic relationships and collinearity analysis indicated close evolutionary relationships between *BrNRT2s* and *AtNRT2s*. Therefore, we could predict the gene functions of *BrNRT2s* based on the gene functions of *AtNRT2s*. *BrNRT2.1*, *BrNRT2.3*, and *BrNRT2.8* only contained one intron; other *BrNRT2* genes had more introns. Previous studies have suggested that the expression levels of genes with fewer introns can rapidly change in response to stress [46]. Moreover, conserved domain and motif analyses were carried out to clarify the relationship between the various members and their potential functions. All *NRT2* family members have PLN00028 domain, indicating that the *NRT2* subfamily might function through the PLN00028 domain. Motif 1, Motif 2, Motif 4, and Motif 5 are present in all *BrNRT2* members, suggesting that they play an essential role in mediating nitrate transport.

Promoters are the regulatory centers of gene transcription, and an in-depth study of promoters can help clarify the mechanisms underlying the regulation of gene transcription. Analysis of the *cis*-elements in the promoter regions of *BrNRT2* genes revealed abundant light-responsive elements, suggesting that the expression of *BrNRT2* genes might be closely related to the regulation of light. In addition, many phytohormone-responsive elements and stress-responsive elements have been identified, suggesting that *BrNRT2* genes may be involved in growth and developmental activities and stress responses in *B. rapa*. GO analysis of the *BrNRT2* genes revealed that the enriched terms in BP were mainly related to nitrate transport, cellular response to nitrate, and nitrate assimilation. Analysis of tissue-specific data showed that *BrNRT2* genes were mainly expressed in the roots, which highlights their importance in nitrate uptake and transport.

An increasing number of studies have shown that nitrogen uptake, transport, and assimilation are related to drought and salt tolerance in plants [27,29]. Therefore, the expression of *BrNRT2* genes has been observed under drought and salt stress conditions. The qRT-PCR results showed that the relative expression of *BrNRT2.3* was significantly upregulated after drought stress. Gene structure analysis showed that *BrNRT2.3* contains only one intron, and the expression level of this gene can change rapidly in response to stress. C*is*-element analysis showed that *BrNRT2.3* contains ABA-responsive elements and MeJA-responsive elements. ABA is a critical hormone that regulates water status and stomatal movement. Under drought conditions, ABA production and accumulation in plant guard cells induce the closure of the stomata to conserve water [47]. MeJA can induce the synthesis of defensive compounds that improve drought resistance by altering various biochemical characteristics of plants, such as increasing the concentration of organic osmoprotectants and oxidase activities [48,49]. *BrNRT2.3* is mainly expressed in the stem and leaf tissues, suggesting that these genes might play a role in the long-distance transport of nitrate from the roots to the aboveground parts. Protein–protein interaction predictions also suggest that the BrNRT2.3 homolog AtNRT2.7 might interact with AtNRT1.1, AtNRT1.5, AtNRT1.6, AtNRT1.7, and AT2G26690. *AtNRT1.1* (*CHL1*) can regulate stomatal opening; a previous study showed that *chl1* mutant plants are drought tolerant because of their ability to reduce water loss [50]. AtNRT1.5 mediates the redistribution of nitrate to the root system and promotes the expression of stress-response-related genes, which enhances salt, drought, and cadmium stress tolerance. The redistribution of nitrate within plants serves as a signal that links diverse stress cues to extensive physiological adjustments, which ultimately enhances their stress tolerance [51]. AtNRT1.6, AtNRT1.7, and AT2G26690 all belong to the PTR2/POT transporter family, and PTR2 in *A. thaliana* is negatively regulated by ABI4 and promotes water uptake during early seed germination [52]. Similarly, BrNRT2.3 may help respond to drought stress by interacting with genes in these families. Therefore, we speculate that when plants initially experience water deficits, their main response is to limit water loss and enhance water uptake; however, when drought persists, *BrNRT2.3* might mediate the rapid response to changes in the external environment and cope with drought stress through the hormone signaling pathway or by interacting with other proteins.

In addition, we found that *BrNRT2.1* and *BrNRT2.8* may play a role in regulating salt stress. Both *BrNRT2.1* and *BrNRT2.8* contain only one intron, and the expression levels of these genes can change rapidly in response to stress. *Cis*-element analysis revealed that *BrNRT2.1* contains ABA-responsive elements and auxin-responsive elements, and *BrNRT2.8* contains MeJA-responsive elements, gibberellin-responsive elements, and ABA-responsive elements; all four hormones play a regulatory role in the response to salt stress. Auxin affects gene expression through a series of functionally distinct transcription factors, including DNA-binding auxin response factors (ARFs). Different ARFs regulate the soluble sugar content and maintain the chlorophyll content to promote the adaptation of plants to salt stress [53]. In rapeseed (*Brassica napus* L. cv. Talaye), the exogenous application of MeJA increased the soluble sugar level, relative water content, and photosynthetic rate to counteract the inhibitory effect of NaCl [54]. In sorghum, the exogenous application of gibberellins (GAs) can alleviate salt-stress-induced cell wall thickening by increasing the cellulose and hemicellulose content of root cells, allowing the rapid entry of water into root cells, and altering the dynamic balance of endogenous hormones in cells, thereby mitigating the effects of salt stress on germination and seedling growth [55]. Under high-salt conditions, ABA can stimulate short-term responses such as stomatal closure, thereby maintaining the water balance and mediating long-term growth responses by regulating the expression of stress-response genes [56]. Tissue-specific analysis showed that *BrNRT2.1* and *BrNRT2.8* are highly expressed in the roots, indicating that these genes may play an important role in the uptake of nitrate from the soil and in coping with root-associated stresses. Both the transcriptome data and qRT-PCR results showed that the relative expression of *BrNRT2.1* and *BrNRT2.8* was significantly elevated under salt stress. The predicted protein interactions also indicated that the BrNRT2.1 and BrNRT2.8 homolog AtNRT2.6 may interact with NRT1.1 and NRT1.8. Previous studies have shown that inhibition of *NRT1.1* expression in plants decreases their root Cl^−^ uptake and reduces NH_4_^+^-conferred salt hypersensitivity [57]. Under stress conditions, NRT1.8 affects the stress tolerance of plants by regulating the partitioning of nitrate between the roots and aboveground parts [58]. These analyses revealed that BrNRT2.1 and BrNRT2.8 might regulate salt stress responses through hormone–protein interactions.

The protein secondary structure analyses revealed that all members of the BrNRT2 family have the highest proportion of α-helixes, and β-turns account for the lowest proportion of secondary structures. Members of the same subgroup exhibit high structural similarity in protein tertiary structure, indicating that they maintained homologous structures during the evolutionary process. The predicted transmembrane structure of NRT2 proteins in *B. rapa* revealed that only the BrNRT2.9 protein exhibits 13 transmembrane domains and can respond rapidly to environmental changes with enhanced nitrate transport. This provides basic information for subsequent in-depth studies of the functions of BrNRT2 proteins.

Based on a comprehensive analysis of sequence features, *cis*-elements, expression profiles, protein interaction, and existing published data, we conclude that *BrNRT2.3* most likely regulates drought stress, whereas *BrNRT2.1* and *BrNRT2.8* most likely regulate salt stress. However, the specific underlying regulatory mechanisms require further investigation.

## Figures and Tables

**Figure 1 genes-14-01564-f001:**
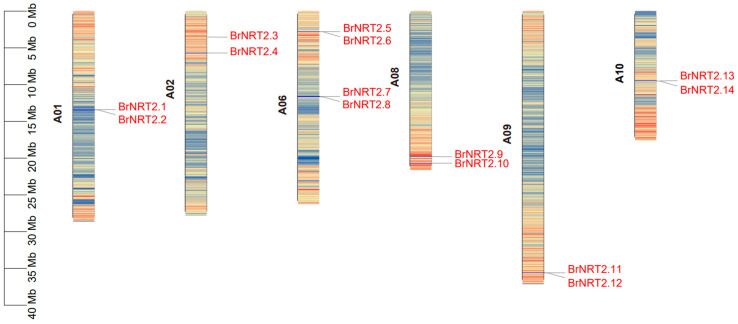
Chromosomal localization of *BrNRT2s*. The chromosomal position of each *NRT2* gene was mapped according to the *B. rapa* genome. The number of chromosomes is listed on the left side of each chromosome. Each chromosome shows gene density.

**Figure 2 genes-14-01564-f002:**
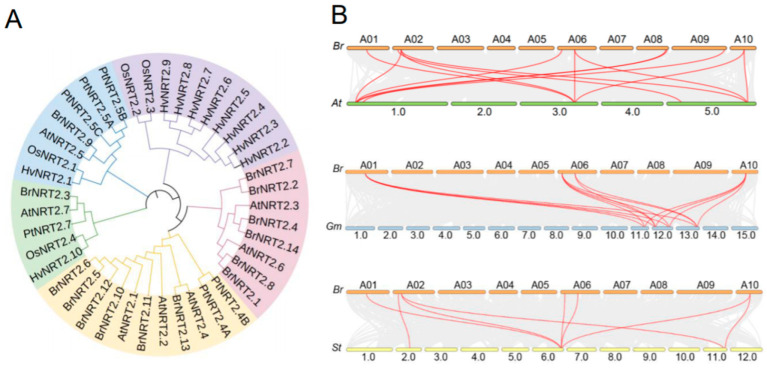
Phylogenetic tree and collinearity analysis of *BrNRT2s.* (**A**) A phylogenetic tree of NRT2 proteins from different plants. Different colors represent different branches; *B. rapa* (*Br*), *A. thaliana* (*At*), *Oryza sativa* (*Os*), *Populus trichocarpa* (*Pt*), *Hordeum vulgare* (*Hv*). (**B**) Collinearity analysis of *BrNRT2* members. The red lines indicate *BrNRT2* gene members with collinearity, and the gray lines indicate other genes; *Glycine max* (*Gm*), *Solanum tuberosum* (*St*).

**Figure 3 genes-14-01564-f003:**
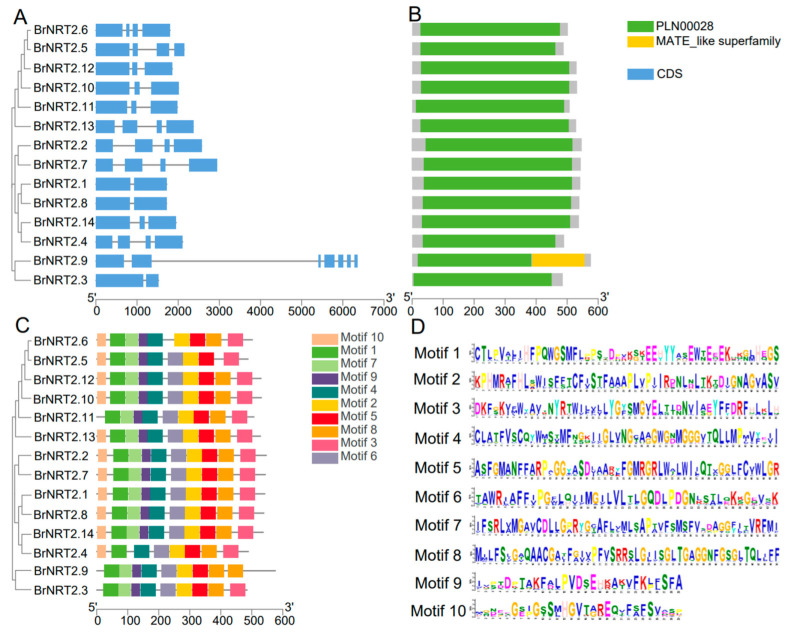
Gene structure and conserved domain analysis of *BrNRT2s*. (**A**) Analysis of gene structure. The introns and exons are shown as black lines and blue boxes, respectively. (**B**) Analysis of conserved domains of *BrNRT2* genes. (**C**) Analysis of the conserved motifs of *BrNRT2* genes. The length of each motif is also shown. (**D**) Motif sequence.

**Figure 4 genes-14-01564-f004:**
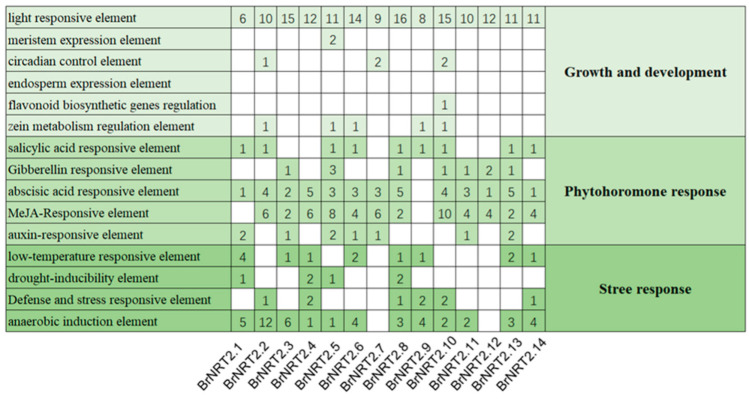
Predicted *cis*-elements in *BrNRT2* promoters. The figure shows the number of *cis*-elements contained in the *BrNRT2* promoters.

**Figure 5 genes-14-01564-f005:**
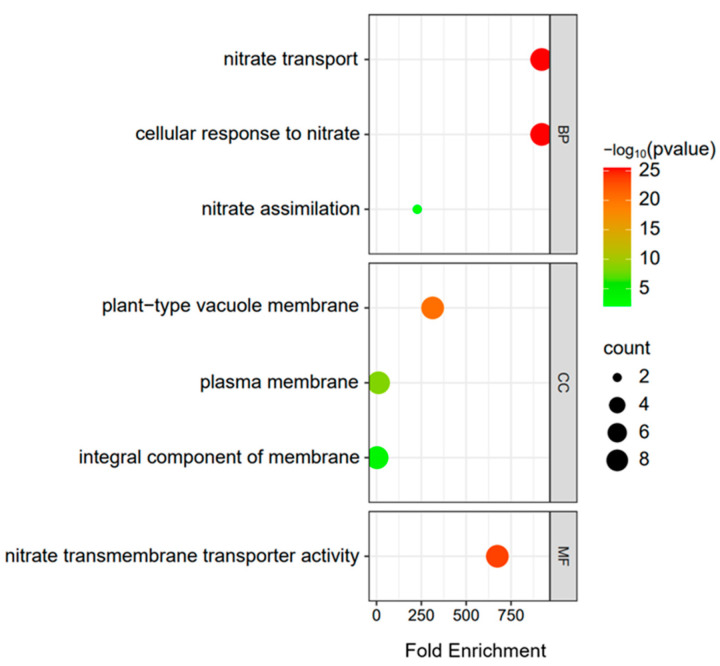
GO enrichment analysis of *BrNRT2s*. The size and color of the dot bubbles indicate the number of genes and *p*-value, respectively.

**Figure 6 genes-14-01564-f006:**
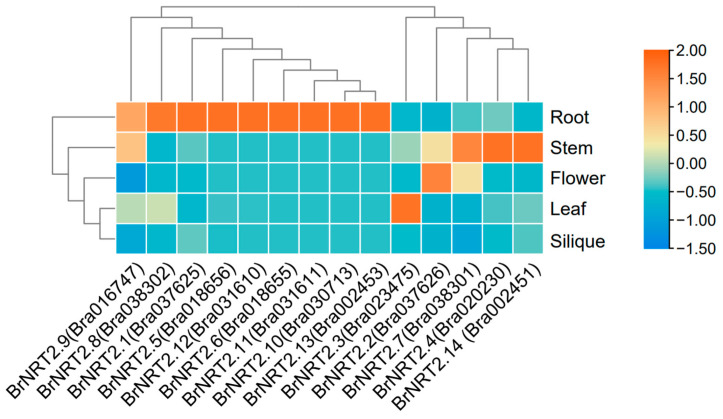
Expression patterns of *BrNRT2s* in different tissues. The abundance of each gene was determined using TPM.

**Figure 7 genes-14-01564-f007:**
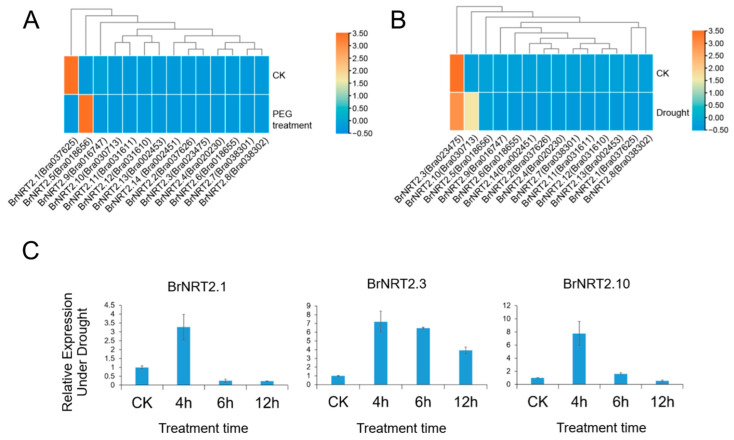
Expression analysis of *BrNRT2s* under drought stress. (**A**) Analysis of transcriptome data of *NRT2* genes under drought stress in drought-sensitive *B. rapa* obtained from the Brassicaceae Database. (**B**) Heatmap of expression patterns of *BrNRT2* genes under drought stress. (**C**) The relative expression levels of *BrNRT2* genes under drought stress were analyzed using qRT-PCR.

**Figure 8 genes-14-01564-f008:**
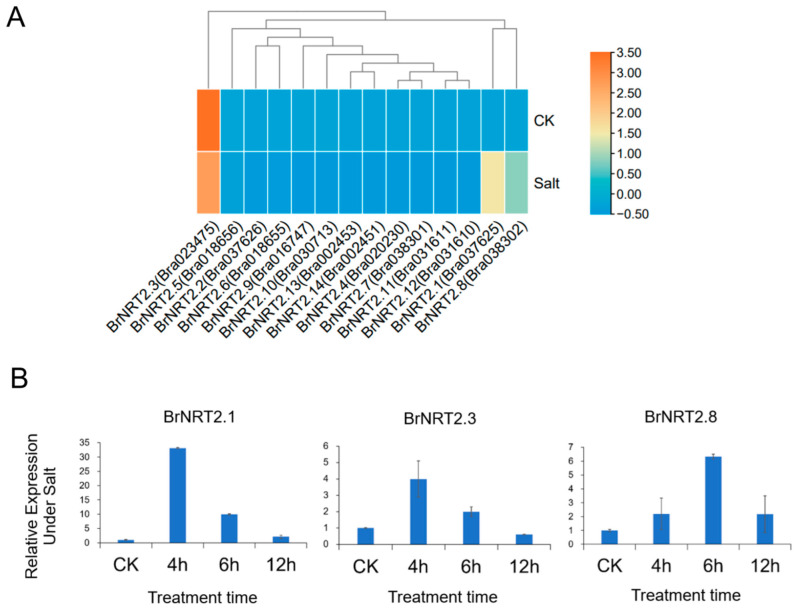
Expression analysis of *BrNRT2s* under salt stress. (**A**) Heatmap of expression patterns of *BrNRT2* genes under salt stress. (**B**) The relative expression levels of *BrNRT2* genes under salt stress were analyzed using qRT-PCR.

**Figure 9 genes-14-01564-f009:**
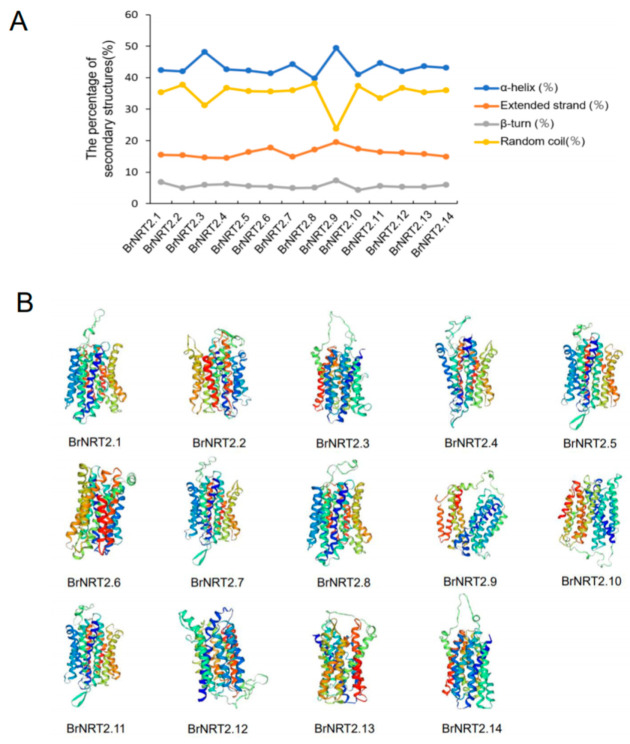
Protein secondary and tertiary structure prediction of BrNRT2s. (**A**) Prediction of the secondary structure of BrNRT2 proteins. Different colors indicate different secondary structures. (**B**) Prediction of the tertiary structures of BrNRT2 proteins. Different colors indicate different subunits.

**Figure 10 genes-14-01564-f010:**
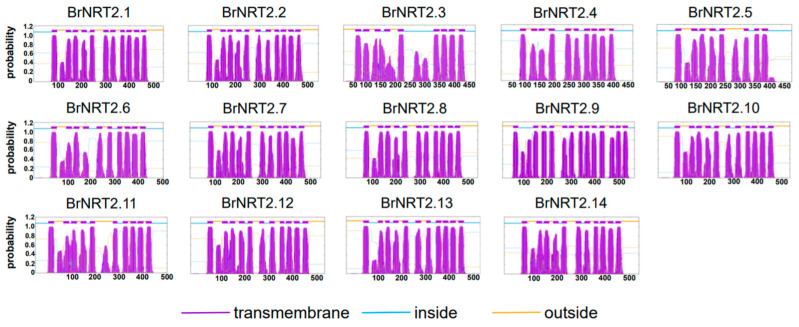
Prediction of BrNRT2 protein transmembrane domains. The gene names are listed above, the *x*-axis shows the length of the protein sequence, and the *y*-axis shows the probability.

**Figure 11 genes-14-01564-f011:**
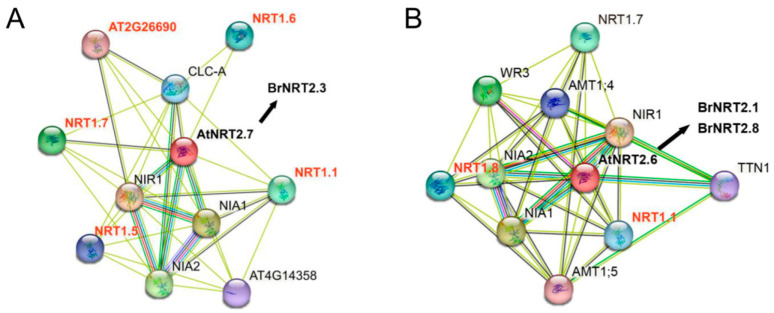
Predicted interaction network of NRT2s. (**A**) AtNRT2.7 (*BrNRT2.3* homologous gene) PPIs. (**B**) AtNRT2.6 (*BrNRT2.1* and *BrNRT2.8* homologous gene) PPIs. The colored nodes represent the query proteins. The filled nodes represent proteins with known or predicted 3D structures. The edges represent protein associations. Different colored lines between the nodes represent the type of evidence for the interaction.

**Table 1 genes-14-01564-t001:** Information of *NRT2* family genes in *B. rapa*.

Gene Name	Gene ID	Chromosome	pI	MW (Da)	Protein Length (aa)	Subcellular Location	*A. thaliana* ID	*A. thaliana* Name
*BrNRT2.1*	*Bra037625*	A01:13400836–13402559	8.85	58460.68	541	Plas	*AT3G45060*	*AtNRT2.6*
*BrNRT2.2*	*Bra037626*	A01:13433817–13436388	6.92	59247.37	546	Plas	*AT3G45060*	*AtNRT2.6*
*BrNRT2.3*	*Bra023475*	A02:3523291–3524816	7.56	52132.17	484	Plas	*AT5G14570*	*AtNRT2.7*
*BrNRT2.4*	*Bra020230*	A02:5693857–5695967	9.15	52839.13	488	Plas	*AT5G60780*	*AtNRT2.3*
*BrNRT2.5*	*Bra018656*	A06:2786900–2789045	9.07	53219.75	487	Plas	*AT1G08090*	*AtNRT2.1*
*BrNRT2.6*	*Bra018655*	A06:2790414–2792217	8.91	54448.02	501	Plas	*AT1G08090*	*AtNRT2.1*
*BrNRT2.* *7*	*Bra038301*	A06:11608821–11611760	6.89	59090.30	543	Plas	*AT3G45060*	*AtNRT2.6*
*BrNRT2.8*	*Bra038302*	A06:11662812–11664531	9.14	58304.72	538	Plas	*AT3G45060*	*AtNRT2.6*
*BrNRT2.9*	*Bra016747*	A08:19790411–19796761	7.55	62619.10	575	Plas	*AT1G12940*	*AtNRT2.5*
*BrNRT2.10*	*Bra030713*	A08:20684138–20686146	8.79	57476.56	530	Plas	*AT1G08090*	*AtNRT2.1*
*BrNRT2.11*	*Bra031611*	A09:35596181–35598163	8.97	54783.89	506	Plas	*AT1G08090*	*AtNRT2.1*
*BrNRT2.12*	*Bra031610*	A09:35601355–35603214	8.99	57381.49	529	Plas	*AT1G08090*	*AtNRT2.1*
*BrNRT2.13*	*Bra002453*	A10:9495122–9497490	8.90	57606.93	527	Plas	*AT5G60770*	*AtNRT2.4*
*BrNRT2.14*	*Bra002451*	A10:9510040–9511991	9.14	58213.61	536	Plas	*AT5G60780*	*AtNRT2.3*

## Data Availability

Data are contained within the article/Appendix A.

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
