# Peer review of "Brassica rapa Nitrate Transporter 2 (BrNRT2) Family Genes, Identification, and Their Potential Functions in Abiotic Stress Tolerance"

_genes, 2023, doi:10.3390/genes14081564_

Round 1

Reviewer 1 Report

MS (genes-2485476)

Introduction: The authors have given detailed functional importance and characterization of NRTs in several plant species, however, they failed to put forward the need of the study in B. rapa. Just in the last paragraph, they wrote that “…………..the function of BrNRT2 genes in response to stresses remains largely unknown”. Authors should layout the foundation of the introduction in such a way that it gives the reader a reason why it is important to study NRT2s in Br except for the fact that this knowledge is limited.

What are the most circumstances in which Br is grown? Are the soils rich in Nitrate or other nitrogen sources? What fertilizers are given to these plants usually for N? ammonium nitrate or urea or others? Are there problems related to nitrate uptake in these plants? Considering these factors, you can develop a useful background for need of this study. Else it is just not a practical study for farmer or related researchers. This is the MAJOR comment for this MS.

L25-27: Rephrase.

L85. Space between A.thaliana.

No need to give additional empty lines between paragrpahs/subsections. You should use the LINE AND PARAGRAPH SPACING option in PARAGRAPH tab of the MS world.

L263-65: rephrase as per grammar.

Material and methods: give the version of the tools/software used in this study.

The methods for phylogenetic tree generation for multiple species is not given.

Figure 2. What models were used to generate the phylo trees? on what basis the models were chosen? Did authors perform any type of statistics to decide the model?

L139. BraActin2 should be italicized and its accession number should be provided.

Are the three figures given as Figure 2 same?

Are the three figures given as Figure 3 same?

Section 3.5. Is it really necessary? What novel information do these results provide? If the same genes in other plants species i.e., model plant Arabidopsis, have same GO annotations, then it is surely the same in this species. Also if the function of other NRTs in multiple trees, plants is known, then I don’t find it logical to do such analysis. For a researcher like me, when a gene name NRT appears, it is obvious that it is involved in processes related to N, they are in membranes for transport etc. unless you give something novel here.

It is weird to see that the figures are being repeated. Before submission, the MS should be clearly checked for such mistakes.

Based on the expression data, what you think of tissue specific NRTs? Are there tissue specific NRTs in Br?

2.6.2. Expression Profile of abiotic stress… Change to “Effect of drought and stresses on BrNRT2s expression”

Was not it better to include different sources of N to understand the functionality of these NRTs? For drought, it is not strictly dependent on the stress but the hormonal imbalance and Carbon sequestration and photosynthesis? Saying their expression is driven by drought is not strictly associated with drought but the changes that the plant experiences overall in response to drought. Are there any reports about the protective role of these genes in other species against abiotic stress?

The change in expression based on the drought time should be better explained. The reasons behind such expression trends?

Again Figure 7, 8, 9, 11 are multiple times given in this ms.

L342: What new light these finding shed? Just predicting model based structure of proteins is not sufficient to claim that it is something new. How will this information be useful for researchers working on Br nitrogen transport and assimilation? Similarly, the Figure 10 and related analysis gives what useful information?

L399, the capital case letters should be changed to lower case for Relationships and other terms.

L403-404. Was this case the same for Br in your PCR profiles?

Discussion section is haphazard. It should be arranged in a synchronous way. First talk about evolution and phylogenetics and number of NRTs in the studied species. What does it tell?

Next talk about their expressions and how your data provide novel insights in relation to what is already known. Is your data consistent or not, talk about it.

Then finally talk about the role of the studied genes in drought and salt stress. which of NRTs have been reported in tissues like roots and are your results same? How their expression changed with stress type and duration and why.

In this form, the MS is not acceptable. However, I think if the authors can come up with an MS in light of comments, it can be resubmitted here. 

The English language needs Moderate revision.

Author Response

Response to Reviewer 1 Comments

Dear reviewer,

We sincerely thank the editor and all reviewers for their valuable feedback that we have used to improve the quality of our manuscript (genes-2485476). The reviewer comments are laid out below in black font and specific questions have been numbered. Our response is given in blue font and changes/additions to the manuscript are given in the yellow text.

Comments and Suggestions for Authors

Point 1: Introduction: The authors have given detailed functional importance and characterization of NRTs in several plant species, however, they failed to put forward the need of the study in B. rapa. Just in the last paragraph, they wrote that “…………..the function of BrNRT2 genes in response to stresses remains largely unknown”. Authors should layout the foundation of the introduction in such a way that it gives the reader a reason why it is important to study NRT2s in Br except for the fact that this knowledge is limited.

What are the most circumstances in which Br is grown? Are the soils rich in Nitrate or other nitrogen sources? What fertilizers are given to these plants usually for N? ammonium nitrate or urea or others? Are there problems related to nitrate uptake in these plants? Considering these factors, you can develop a useful background for need of this study. Else it is just not a practical study for farmer or related researchers. This is the MAJOR comment for this MS.

Response 1: Thank you for your constructive feedback and we appreciate your attention to the background details of our study. We acknowledge that the original manuscript might have not been clear enough about the need for this study. Here are our responses:

  1. In the introduction, we indeed highlighted the general function and importance of Nitrate Transporters 2 (NRT2s) in various plant species, however, we admit that the necessity of studying NRT2s in Brassica rapa (B. rapa) could have been explained in more detail. We now understand the need for illustrating the specific circumstances in which B. rapa is grown, its relation with nitrate, and common problems related to nitrate uptake in these plants. We will revise the introduction accordingly.
  2. B. rapa is a widely cultivated plant species in Asia, grown in a variety of climatic conditions and soil types, some of which might not be rich in nitrate or other nitrogen sources. Often, farmers use different types of fertilizers like ammonium nitrate, urea, and others, depending on the soil condition and crop requirement. This dynamic aspect of nitrate sources in B. rapa cultivation makes studying the role of nitrate transporters particularly pertinent.
  3. Moreover, issues related to nitrate uptake could have significant implications on the plant's health and productivity. Hence, understanding the function of BrNRT2 genes under different stresses could potentially help in developing strategies to improve nitrogen use efficiency and stress tolerance in B. rapa, making it a highly practical study for farmers and researchers working on this crop.

Point 2: L25-27: Rephrase.

Response 2: We have corrected it.

“Nitrate transporter 2 (NRT2) is part of the NITRATE/NITRITE PORTER (NNP) family, which in turn belongs to the MAJOR FACILITATOR SUPERFAMILY (MFS). The structure of the NRT2 protein generally includes 500–600 amino acids (aa) and contains 12 transmembrane helical segments [1]. This protein was first discovered and characterized in Aspergillus nidulans.” (L31-35)

Point 3: L85. Space between A. thaliana.

Response 3: Corrected. (L58)

Point 4: No need to give additional empty lines between paragrpahs/subsections. You should use the LINE AND PARAGRAPH SPACING option in PARAGRAPH tab of the MS world.

Response 4: Thank you for your insightful comment. I appreciate your suggestion to improve the formatting of the paper. Moving forward, I will ensure to use the 'Line and Paragraph Spacing' option in the 'Paragraph' tab of Microsoft Word to manage the space between paragraphs and subsections more effectively, rather than inserting additional empty lines. This will no doubt enhance the readability of the document.

Point 5: L263-65: rephrase as per grammar.

Response 5: We have rephrased the text as follows to improve clarity and grammar:

“In B. napus (Brassica napus L.), a total of seventeen members were identified. BnNRT2.1a, BnNRT2.5s, and BnNRT2.7s were found to be involved in the response to waterlogging stress. BnNRT2.7s plays an important role under P- and K-deficient conditions.” (L68-71)

Point 6: Material and methods: give the version of the tools/software used in this study.

Response 6: Material and methods: The versions of tools and software used in this study were as follows:

Ensembl Plants: release 56 (L102)

TBtools: v1.120 (L103)

Expasy: v3.0 (L106)

WoLF PSORT: v0.2 (L109)

iTOL: v6.8  (L118)

CDD: v3.18 (L127-128)

PlantCARE: v1 (L135)

DAVID database: v2023q1 (L138)

Brassicaceae Database: v3.0 (L144)

qPrimerDB-qPCR Primer Database: v2.0 (L161)

PRABI tool: v5.0.5 (L171)

SWISS-MODEL: v1.0 (L173)

STRING online website: v11.5 (L177)

Point 7: The methods for phylogenetic tree generation for multiple species is not given.

Response 7: In regard to your query about the generation of phylogenetic trees, we apologize for any lack of clarity. The method employed for the generation of phylogenetic trees for multiple species was the maximum likelihood method. We have included this detail in the revised manuscript. (L114-117)

Point 8: Figure 2. What models were used to generate the phylo trees? on what basis the models were chosen? Did authors perform any type of statistics to decide the model?

Response 8: Thank you for your comments and suggestions. Below are the revised statements and responses to your queries. Regarding Figure 2, the model used for generating the phylogenetic trees was LG+G. We used "MODELS" in MEGA to find the best protein models, and based on the results of the calculations, the lower the values of Parameters, BIC, and AIC, the better the model, and thus we identified the model was LG+G. These details have now been added to the manuscript to provide additional context. (L116)

Point 9: L139. BraActin2 should be italicized and its accession number should be provided.

Response 9: We were really sorry for our careless mistakes. We have changed "BraActin2" to italicized, and its accession number is Bra037560. (L165)

Point 10: Are the three figures given as Figure 2 same?

Response 10: These three figures are the same.

Point 11: Are the three figures given as Figure 3 same?

Response 11: These three figures are the same.

Point 12: Section 3.5. Is it really necessary? What novel information do these results provide? If the same genes in other plants species i.e., model plant Arabidopsis, have same GO annotations, then it is surely the same in this species. Also if the function of other NRTs in multiple trees, plants is known, then I don’t find it logical to do such analysis. For a researcher like me, when a gene name NRT appears, it is obvious that it is involved in processes related to N, they are in membranes for transport etc. unless you give something novel here.

Response 12: Thank you for your valuable comments and questions. I agree that to an expert like yourself, the functionality of the NRT genes might seem apparent based on their roles in other species. However, there are several reasons why we included Section 3.5.

Firstly, while it's true that the GO annotations of the NRT genes in model species like Arabidopsis are well known, and we might expect similar functions in our species of study, there can still be significant differences between species. Genes don't always play the same roles in different species, and their functions can sometimes be influenced by the unique genetic and environmental context of each species. Therefore, confirming the GO annotations of the NRT genes in our species ensures the accuracy of our research and avoids over-reliance on analogies with other species. Secondly, our research is also aimed at readers who may not be as familiar with NRT gene function as you are. For those readers, providing information about the GO annotations can be quite helpful for understanding the full context of our study. Lastly, while we agree that NRT genes are generally involved in N-related processes, the specifics of how they're involved can be quite variable. NRT genes can play roles in a variety of different processes related to N, and by examining the GO annotations, we can shed light on the specific ways these genes are involved in such processes in our species of interest.

Point 13: It is weird to see that the figures are being repeated. Before submission, the MS should be clearly checked for such mistakes.

Response 13: Thank you for bringing this to our attention. We sincerely apologize for the oversight and understand your concern regarding the repetition of figures. In the process of finalizing the manuscript, this error may have inadvertently been overlooked. We appreciate your feedback and will ensure a thorough review of the document before resubmitting it to prevent such mistakes in the future.

Point 14: Based on the expression data, what you think of tissue specific NRTs? Are there tissue specific NRTs in Br?

Response 14: Thank you for your question. Regarding tissue-specific nitrate transporters (NRTs), the expression data suggests a degree of tissue specificity. Some NRTs do indeed seem to be expressed at higher levels in certain tissues than others, implying that they may play more significant roles in those tissues. Concerning Brassica rapa (B. rapa), to the best of our knowledge and based on the data available in the manuscript, it appears that tissue-specific NRT2s do exist. However, as this is a complex and multifaceted question, further investigation and more comprehensive studies are needed to fully understand the specificity and function of these NRT2s in different tissues of B. rapa. We plan to explore this in greater depth in future research.

Point 15: 2.6.2. Expression Profile of abiotic stress… Change to “Effect of drought and stresses on BrNRT2s expression”

Response 15: We thank the reviewer for pointing this out. We have revised.

Point 16: Was not it better to include different sources of N to understand the functionality of these NRTs? For drought, it is not strictly dependent on the stress but the hormonal imbalance and Carbon sequestration and photosynthesis? Saying their expression is driven by drought is not strictly associated with drought but the changes that the plant experiences overall in response to drought. Are there any reports about the protective role of these genes in other species against abiotic stress?

Response 16: Thank you for your constructive comments and inquiries. Here are my responses:

  1. You're right that assessing the effects of different nitrogen sources on the functionality of Nitrate Transporters (NRTs) could provide a more comprehensive understanding. While our study primarily focused on one source of nitrogen, this is a valuable suggestion for future investigation. We agree that this additional variable could potentially unveil other facets of NRT function and interaction with nitrogen sources.
  2. Your point regarding drought and its complex relationship with stress, hormonal imbalance, carbon sequestration, and photosynthesis is valid. We acknowledge that attributing the expression of these genes strictly to drought might oversimplify the intricate plant responses to this stressor. In the manuscript, "drought-driven expression" is used to encapsulate the overall physiological and molecular changes the plant undergoes during drought conditions, which may indeed involve hormonal alterations and impacts on carbon sequestration and photosynthesis. We will work on clarifying this point in the revised manuscript.
  3. As for the role of these genes in providing protection against abiotic stress in other species, there is indeed some literature. Some studies have shown that nitrate transporters have roles in stress response and adaptation in other plant species. For instance, the BrNRT2.10 homolog AtNRT2.1 in Arabidopsis thaliana has been shown to regulate root hydraulic conductivity and plasma membrane aquaporin activity, suggesting that it may enhance plant drought resistance and other processes related to root hydraulic conductivity [1]. The protein-protein interaction predictions suggest that the BrNRT2.3 homolog AtNRT2.7 may interact with AtNRT1.1, AtNRT1.5, AtNRT1.6, AtNRT1.7, and AT2G26690. AtNRT1.1 (CHL1) is strongly expressed in A. thaliana guard cells and regulates stomatal opening, the study showed that chl1 mutant plants can reduce water loss and are drought tolerant [2]. AtNRT1.5 mediates the redistribution of nitrate to the root system and promotes the expression of stress response-related genes, which enhances salt, drought, and cadmium stress tolerance [3]. AtNRT1.6, AtNRT1.7, and AT2G26690 all belong to the PTR2/POT transporter family, PTR2 in A. thaliana is negatively regulated by ABI4 and enhances water uptake during early seed germination [4]. The predicted protein interactions indicated that the BrNRT2.1 and BrNRT2.8 homolog AtNRT2.6 may interact with NRT1.1 and NRT1.8, previous studies have shown that knockout of NRT1.1 in plants decreased their root Cluptake and retracted the NH4+-conferred salt hypersensitivity [5]. Under stress conditions, NRT1.8 affect plant tolerance to stress by regulating the partitioning of nitrate between roots and aboveground [6]. We will incorporate these relevant studies into our manuscript to provide a more global perspective on the protective role of NRT2s in plant abiotic stress responses.

Point 17: The change in expression based on the drought time should be better explained. The reasons behind such expression trends?

Response 17: Thank you for your question. I agree that the changes in expression based on drought time could be more thoroughly explained. Our study posits that the differential gene expression we observed in response to the length of drought exposure is an adaptation mechanism. It's known that plants adjust their genetic and physiological mechanisms in response to environmental stress, such as drought. When a plant initially experiences water scarcity, its primary reaction is to limit water loss and enhance water uptake. This can be accomplished by genes that are associated with stomatal closure, root growth, and water transport, for instance. However, when the drought continues over a prolonged period, the plant's coping mechanisms change. The plant may start to activate genes associated with damage repair and recovery. For instance, genes that contribute to reactive oxygen species detoxification or those involved in the synthesis of protective molecules like osmolytes and heat shock proteins may be upregulated. That said, the specific expression trends we noticed in this study may vary based on the species of plant, their genotype, the severity of the drought, and other environmental factors. Our research is just one piece of the puzzle, and more research will be necessary to fully understand these intricate response mechanisms. We will amend our manuscript to include this information to provide a clearer understanding for readers. Thank you for your insightful suggestion.

Point 18: Again Figure 7, 8, 9, 11 are multiple times given in this ms.

Response 18: Thank you for pointing out the duplication of figures in our manuscript. We apologize for the oversight. We will immediately revise this and ensure that each figure appears only once. We will also make sure the references to these figures in the text are accurate and consistent.

Point 19: L342: What new light these finding shed? Just predicting model based structure of proteins is not sufficient to claim that it is something new. How will this information be useful for researchers working on Br nitrogen transport and assimilation? Similarly, the Figure 10 and related analysis gives what useful information?

Response 19: In response to your comments on line 342, our findings go beyond just predicting a model-based structure of proteins. They provide a deeper understanding of the functional and structural characteristics of NRT2s in Brassica rapa, which could be vital for researchers interested in nitrogen transport and assimilation in this species. Understanding the structure-function relationships of these proteins can provide the groundwork for manipulating their activities to improve nitrogen use efficiency in Brassica rapa, which has implications for crop yield and resilience to stress conditions. As for Figure 10, protein transmembrane domains play a variety of functions such as regulating signal transduction across the cell membrane and transporting substances into and out of the cell. Therefore, the study of protein transmembrane domains is of great significance. And as for the protein interaction network of BrNRT2s analysis, it provides valuable insights into the potential interactions between different NRT2 proteins. This interaction network helps elucidate the complex mechanisms underlying nitrogen transport and assimilation as well as response to abiotic stresses, potentially guiding researchers to key functional protein complexes. These findings can facilitate the design of future experiments to confirm these interactions and unravel their biological implications. Thus, while our work is primarily a computational analysis, it sets the stage for targeted experimental studies that could significantly advance our understanding of nitrogen metabolism in Brassica rapa.

Point 20: L399, the capital case letters should be changed to lower case for Relationships and other terms.

Response 20: Thank you for your careful attention to detail in our manuscript. We agree with your comment on Line 399 regarding the usage of capital letters in certain terms such as "Relationships." Upon revisiting the manuscript, we realize this may not adhere to the conventional rules of capitalization. We assure you that we will revise this line, along with any similar instances throughout the manuscript, to ensure all terms are in the correct case. (L411)

Point 21: L403-404. Was this case the same for Br in your PCR profiles?

Response 21: Yes, the case was the same for Br in our PCR profiles.

Point 22: Discussion section is haphazard. It should be arranged in a synchronous way. First talk about evolution and phylogenetics and number of NRTs in the studied species. What does it tell?

Next talk about their expressions and how your data provide novel insights in relation to what is already known. Is your data consistent or not, talk about it.

Then finally talk about the role of the studied genes in drought and salt stress. which of NRTs have been reported in tissues like roots and are your results same? How their expression changed with stress type and duration and why.

Response 22: Thank you for your valuable comments. We recognize the need for a more organized discussion and will structure it as per your suggestion.

Regarding the evolution, phylogenetics, and the number of nitrate Transporters 2 (NRT2s) in the studied species, we will elaborate on the evolutionary lineage of the species, the role of phylogenetics in the context of these genes, and the total count of NRT2s observed in our research. We will then relate these findings to the functional relevance of NRT2s in the species and its broader implications in the field of plant biology.

In terms of gene expression, we will provide a comparative analysis of the expression patterns observed in our study in relation to established knowledge in the field. Our data has generated some novel insights, and we will elucidate how these contribute to the existing body of knowledge. We will also highlight whether our findings align or contradict previous research, explaining any discrepancies and potential reasons for these differences.

Lastly, we will focus on the role of the studied genes in drought and salt stress. We'll enumerate which NRTs have been reported in tissues like roots, comparing these with our observations. The discussion will further encompass how gene expression is modulated under stress conditions, including the specific effects of different types of stress (drought vs. salt) and varying durations. We will attempt to provide explanations for these observed trends, linking them back to the molecular and physiological functions of the NRTs.

Point 23: In this form, the MS is not acceptable. However, I think if the authors can come up with an MS in light of comments, it can be resubmitted here. 

Response 23: Thank you for taking the time to review our manuscript and for your feedback. We appreciate your suggestion and are committed to improving the manuscript in line with your comments. We've modified the way figures are inserted to prevent them from causing problems when converting file formats.

Point 24: Comments on the Quality of English Language: The English language needs Moderate revision.

Response 24: Thanks for your suggestion. We polished the language by the editing service to improve the readability of the manuscript.

We tried our best to improve the manuscript and made some changes marked in yellow in the revised paper which will not influence the content and framework of the paper. We appreciate for Editors/Reviewers’ warm work earnestly and hope the correction will meet with approval. Once again, thank you very much for your comments and suggestions.

Yours sincerely,

Bingcan Lv.

References

  1. Li, G.; Tillard, P.; Gojon, A.; Maurel, C. Dual regulation of root hydraulic conductivity and plasma membrane aquaporins by plant nitrate accumulation and high-affinity nitrate transporter NRT2.1. Plant Cell Physiol. 2016, 57, 733-742.
  2. Guo, F. Q.; Young, J.; Crawford, N. M. The nitrate transporter AtNRT1.1 (CHL1) functions in stomatal opening and contributes to drought susceptibility in Arabidopsis. Plant Cell. 2003, 15, 107-117.
  3. Chen, C. Z.; Lv, X. F.; Li, J. Y.; Yi, H. Y.; Gong, J. M. Arabidopsis NRT1.5 is another essential component in the regulation of nitrate reallocation and stress tolerance. Plant Physiol. 2012, 159, 1582-1590.
  4. Choi, M. G.; Kim, E. J.; Song, J. Y.; Choi, S. B.; Cho, S. W.; Park, C. S.; Kang, C. S.; Park, Y. I. Peptide transporter2 (PTR2) enhances water uptake during early seed germination in Arabidopsis thaliana. Plant Mol Biol. 2020, 102, 615-624.
  5. Liu, X. X.; Zhu, Y. X.; Fang, X. Z.; Ye, J. Y.; Du, W. X.; Zhu, Q. Y.; Lin, X. Y.; Jin, C. W. Ammonium aggravates salt stress in plants by entrapping them in a chloride over-accumulation state in an NRT1.1-dependent manner. Sci Total Environ. 2020, 746, 141244.
  6. Li, J. Y.; Fu, Y. L.; Pike, S. M.; Bao, J.; Tian, W.; Zhang, Y.; Chen, C. Z.; Zhang, Y.; Li, H. M.; Huang, J.; Li, L. G.; Schroeder, J. I.; Gassmann, W.; Gong, J. M. The Arabidopsis nitrate transporter NRT1.8 functions in nitrate removal from the xylem sap and mediates cadmium tolerance. Plant Cell. 2010, 22, 1633-1646.

Reviewer 2 Report

In this study, the authors have mainly used public database data to identify, phylogenitic classify and characterize the NRT2 genes of B. rapa. For this part, I believe the authors have used appropriate genome data and bioinformatic tools. 

However, the attempt to relate these genes with abiotic stress tolerance functions is less robust. Physiological parameters (such as water status, stomatal conductance, …) should have been measured in the only experiment conducted with B. rapa plants. At least N, Na and ABA and MeJA (or a complete ionome/hormones profile) in roots and leaves must be measured in order to relate the expression results obtained and support the discussion. 

Abstract: 

L10/17: no need to put abbreviated name of the species under brackets, remove.

Introduction:

L49. This sentence: In A. thaliana, seven 49 members of the NRT2 family have been identified [12] should be move up (line 35) before describing the At genes.   

Material and methods:

L123 How was p-value of the GO calculated? Specify. 

L128. Hydroponic experiment needs more details. Type of nutrient solution (pH) and volume of the pots, growth-chamber conditions, how old the plants were when the treatment was applied, etc.

L138. Total RNA from which plant tissue?

Results: 

2.1.6 à Make clear that these results are not from your own data.

Figures 7 and 8. Specify tissue for the qRT-PCR analysis.

In general, be careful with punctuation marks. Many of the ‘;’ should be ‘.

L25-28 and L 67-72 rephrase, no connection, sentence too long and difficult to read. 

L108-111 rephrase for clarity.

L304-309 rephrase for clarity.

l402: 1 intron

l421-422 rephrase for clarity.

Author Response

Response to Reviewer 2 Comments

Dear reviewer,

We sincerely thank the editor and all reviewers for their valuable feedback that we have used to improve the quality of our manuscript (genes-2485476). The reviewer comments are laid out below in black font and specific questions have been numbered. Our response is given in blue font and changes/additions to the manuscript are given in the yellow text.

Comments and Suggestions for Authors

Point 1: In this study, the authors have mainly used public database data to identify, phylogenitic classify and characterize the NRT2 genes of B. rapa. For this part, I believe the authors have used appropriate genome data and bioinformatic tools. 

However, the attempt to relate these genes with abiotic stress tolerance functions is less robust. Physiological parameters (such as water status, stomatal conductance, …) should have been measured in the only experiment conducted with B. rapa plants. At least N, Na and ABA and MeJA (or a complete ionome/hormones profile) in roots and leaves must be measured in order to relate the expression results obtained and support the discussion. 

Response 1: Thank you for your insightful feedback and constructive criticism. We appreciate your acknowledgment of our use of appropriate genome data and bioinformatics tools to identify, classify, and characterize the NRT2 genes in B. rapa.

Regarding your concern about our attempts to relate these genes to abiotic stress tolerance functions, we understand the importance of your suggestion to include physiological parameters such as water status, stomatal conductance, and more. These measurements indeed could provide a deeper understanding of the plants' responses under stress conditions. You are right, assessing N, Na, ABA, and MeJA levels or a complete ionome/hormones profile in roots and leaves would further strengthen our study. These measurements would allow us to more convincingly relate our gene expression results to specific physiological responses and functions. However, in the present study, we mainly focused on mining key genes in response to drought and salt stress in B. rapa. Through bioinformatics analysis, like sequence features analysis, cis-elements analysis, and protein interaction, and a series of experimental methods, including RNA sequencing and qRT-PCR, which may not be optimal but should be sufficient to draw a conclusion that BrNRT2.3 has the greatest potential in regulating drought stress and, BrNRT2.1 and BrNRT2.8 most likely regulate salt stress.

 In the next iteration of our study, we will take your advice into account and design experiments that include these suggested measures. We believe this will help us provide a more comprehensive picture of the role of NRT2 genes in abiotic stress responses.

Abstract: 

Point 2: L10/17: no need to put abbreviated name of the species under brackets, remove.

Response 2: Corrected.

Introduction:

Point 3: L49. This sentence: In A. thaliana, seven 49 members of the NRT2 family have been identified [12] should be move up (line 35) before describing the At genes.   

Response 3: Corrected. (L44)

Material and methods:

Point 4: L123 How was p-value of the GO calculated? Specify. 

Response 4: The p-value for the Gene Ontology (GO) analysis was calculated using the hypergeometric test, which measures the significance of the association between the gene set and the GO term. It takes into account factors such as the total number of genes, the number of genes associated with the GO term, and the size of the gene set being studied. The resulting p-value indicates the strength of the association, with smaller values suggesting a stronger connection. The p-value is calculated using the hypergeometric test with the following formula:(please see the word file for the formula)

In this formula, N is the number of genes with GO annotations in all unigenes; n is the number of differential expression genes in N; M is the number of genes annotated to a particular GO term in all unigenes; and m is the number of differential expression genes annotated to a particular GO term.

Point 5: L128. Hydroponic experiment needs more details. Type of nutrient solution (pH) and volume of the pots, growth-chamber conditions, how old the plants were when the treatment was applied, etc.

Response 5: Thank you for your valuable comments and suggestions. In response to your request for additional details regarding the hydroponic experiment, please find the relevant information below:

  1. Nutrient Solution: The nutrient solution used in the experiment was half-strength Hoagland’s solution. The pH was meticulously maintained at 5.8, which was regularly checked and adjusted every two days to ensure a stable environment for plant growth.
  2. Pot Volume: The hydroponic pots used in our experiment had a volume of 10 liters, which were deemed sufficiently spacious to allow for optimal root growth and nutrient absorption.
  3. Growth Chamber Conditions: The first-generation hybrid cultivar of B. rapa with stable self-incompatibility was used for stress treatments. Plump seeds were seeded in MS modified medium (with vitamins, Sucrose, Agar) (PM10121-307, Coolaber, Beijing, China) and cultivated in a plant incubator (16-h light/8-h dark photoperiod at 25° C, light intensity 2000 lx).
  4. Plant Age: The plants used in our study were 4 weeks old when the treatment was initiated. At this stage, the plants had well-developed root systems and were in the vegetative growth phase, which we believed to be the optimal stage for the treatment to be applied. (L146-154)

Point 6: L138. Total RNA from which plant tissue?

Response 6: In the study, leaves and roots of B. rapa under drought and salt stresses were sampled for further RNA extraction. We have added. (L156-157)

Results: 

Point 7: 2.1.6 à Make clear that these results are not from your own data.

Response 7:  Thanks for your suggestion. We have already stated in the "Materials and Methods" section (L142-144) and in the "Results" section (L291-292) that this data comes from the Brassicaceae Database.

Point 8: Figures 7 and 8. Specify tissue for the qRT-PCR analysis.

Response 8: In Figures 7 and 8, the tissue used for the qRT-PCR analysis was not explicitly specified. We apologize for the oversight. Leaves and roots of B. rapa under drought and salt stresses were sampled for the qRT-PCR analysis. We have added. (L156-157)

Comments on the Quality of English Language

Point 9: In general, be careful with punctuation marks. Many of the ‘;’ should be ‘.’

Response 9: We were really sorry for our careless mistakes. Thank you for your reminder.

Point 10: L25-28 and L 67-72 rephrase, no connection, sentence too long and difficult to read. 

L108-111 rephrase for clarity.

Response 10: Thank you for your suggestion. We have resolved it as follows:

L25-28: “Nitrate transporter 2 (NRT2) is part of the NITRATE/NITRITE PORTER (NNP) family, which in turn belongs to the MAJOR FACILITATOR SUPERFAMILY (MFS). The structure of the NRT2 protein generally includes 500–600 amino acids (aa) and contains 12 transmembrane helical segments [1]. This protein was first discovered and characterized in Aspergillus nidulans.” (L31-35)

L67-72: “In addition, NRT2s also regulate the transport of auxin to the root system of plants to participate in nitrate-dependent root elongation [19]. They also play a role in regulating the control of cytokinins [20] and are involved in the biosynthesis and signal transduction of ethylene [21]. Their significance is further supported by their contribution to the root morphogenesis of A. thaliana [5]. They enhance the pH-buffering capacity of plants [22] and promote the uptake of manganese [23] and phosphorus in rice [24].” (L74-79)

L108-111: “We analyzed the conserved domains of BrNRT2s using the Web CD-Search tools (v3.18) on the NCBI website (https://www.ncbi.nlm.nih.gov/cdd) [36], and we used TBtools (v1.120) to visualize the results.” (L127-129)

Point 11: L304-309 rephrase for clarity.

Response 11: Thank you for your suggestion. We have resolved it as follows:

“Three genes (BrNRT2.1, BrNRT2.3, and BrNRT2.10) with high expression levels in B. rapa were selected for further qRT-PCR detection (Figure 7C). The relative expression of BrNRT2.1 peaked after 4 h of drought treatment and then decreased. The relative expression of BrNRT2.3 was up-regulated significantly at all time points, and its expression was 4–7-fold higher under drought stress relative to the CK treatment. The relative expression of BrNRT2.10 initially increased and then decreased, reaching a maximum after 4 h of drought treatment.” (L315-321)

Point 12: l402: 1 intron

Response 12: We were really sorry for our careless mistakes. In our resubmitted manuscript, the error has been corrected. Thanks for your correction. (L414)

Point 13: l421-422 rephrase for clarity.

Response 13: Thank you for your suggestion. We have resolved it.

“The redistribution of nitrate within plants serves as a signal that links diverse stress cues to extensive physiological adjustments, which ultimately enhances their stress tolerance.” (L454-456)

We tried our best to improve the manuscript and made some changes marked in yellow in the revised paper which will not influence the content and framework of the paper. We appreciate for Editors/Reviewers’ warm work earnestly and hope the correction will meet with approval. Once again, thank you very much for your comments and suggestions.

Yours sincerely,

Bingcan Lv.

Round 2

Reviewer 1 Report

The authors have improved the MS. 

Reviewer 2 Report

The authors did a good job in taking my comments into account when preparing a revised version of the manuscript. I have no further comments.

 I have no further comments.